# OpenReview forum: "ResearchCodeBench: Benchmarking LLMs on Implementing Novel Machine Learning Research Code"
_NeurIPS.cc/2025/Datasets_and_Benchmarks_Track — NeurIPS 2025 Datasets and Benchmarks Track spotlight_

### Official Review · Reviewer_HFEz · 2025-06-30

**Rating:** 5
**Confidence:** 3

**Summary:**

This paper introduces ResearchCodeBench, a benchmark designed to evaluate large language models' ability to translate novel machine learning research contributions from papers into executable code. The benchmark consists of 212 coding challenges derived from 20 recent ML papers (2024-2025) across diverse topics including generative modeling, computer vision, and reinforcement learning. The authors evaluate 30+ LLMs and find that even the best models correctly implement less than 40% of the code (using a scaled pass@1 metric). The benchmark includes executable tests co-developed with paper authors to ensure correctness.

**Dataset Code Accessibility:**

Yes

**Ethical Considerations:**

No, there are no or only very minor ethics concerns

**Final Justification:**

My concerns are addressed. Hence I keep my positive rating.

**Limitations Weaknesses:**

1. The absence of human performance data makes it difficult to contextualize the ~40% success rate. While the authors cite cost as a barrier, even a small human study would provide valuable calibration.
2. With only 20 papers and 212 tasks, all from the ML domain, the benchmark's coverage is narrow. This limits its ability to assess LLMs' broader research code generation capabilities.
3. The paper reports only single-run results without error bars or confidence intervals, limiting the reliability of performance comparisons between models.

**Strengths Contributions:**

1. The benchmark tackles the important but underexplored problem of evaluating LLMs on implementing genuinely novel research ideas, rather than reproducing known algorithms. This is highly relevant for understanding LLMs' potential as research assistants.
2. The construction process is well-designed, especially involving collaboration with original paper authors or domain experts to ensure fidelity. The use of executable tests provides objective evaluation criteria, avoiding the pitfalls of subjective LLM-based judging.
3. The careful selection of papers from 2024-2025 (with 13/20 having first commits in 2025) ensures the benchmark tests out-of-distribution generalization, as demonstrated by the performance drop on the contamination-safe subset.
The paper provides thorough analysis including contamination studies, paper reliance analysis, and detailed error categorization showing that 58.6% of failures are functional/semantic rather than syntactic.

---

> ### Author Rebuttal · Authors · 2025-07-31
>
> **Reviewer Comment:**
> The absence of human performance data makes it difficult to contextualize the ~40% success rate. While the authors cite cost as a barrier, even a small human study would provide valuable calibration.
>
> **Author Response:**
>
> Thank you for the helpful suggestion. We agree that human performance data would provide valuable calibration for our ~40% success rate results. During the rebuttal period, we conducted a preliminary human study with one expert participant familiar with the selected research areas.
>
> *Note: The performance data in the following table is also presented in our response to reviewer 6kcd. All success rates reported here use pass@1 metrics (tasks passed / total tasks).*
>
> **Table: Human Expert Performance on ResearchCodeBench (Preliminary Study)**
>
> | Paper | Human Performance | Total Tasks |
> |-------|-------------------|-------------|
> | advantage-alignment (Duque et al., 2025) | 3/5 (60%) | 5 |
> | Diff-Transformer (Ye et al., 2025) | 5/7 (71%) | 7 |
> | DiffusionDPO (Wallace et al., 2024) | 9/9 (100%) | 9 |
> | DyT (Zhu et al., 2025) | 9/9 (100%) | 9 |
> | GPS (Zhang et al., 2025) | 6/6 (100%) | 6 |
> | eomt (Kerssies et al., 2025) | 30/37 (81%) | 37 |
> | hyla (Schug et al., 2025) | 2/3 (67%) | 3 |
> | llm-sci-use (Liang et al., 2024) | 6/15 (40%) | 15 |
> | Tanh-Init (Lee et al., 2024) | 4/4 (100%) | 4 |
> | minp (Nguyen et al., 2024) | 5/7 (71%) | 7 |
> | **Overall** | **79/102 (77.5%)** | **102** |
>
> Comparing this with LLM performance on the same 10 papers reveals important insights:
>
> **Comparative Performance (Same 10 Papers, 102 Tasks, Standard Pass@1):**
> - **Human expert**: 77.5% (79/102)
> - **Best LLM** (Gemini-2.5-Pro-Preview-05-06): 68.6% (70/102)
> - **Performance gap**: 8.9 percentage points
>
> **Table: Top 5 LLMs vs Human Performance (Same 10 Papers)**
>
> | Model | Performance | Gap vs Human |
> |-------|-------------|--------------|
> | Human Expert | 77.5% (79/102) | - |
> | Gemini-2.5-Pro-Preview-05-06 | 68.6% (70/102) | -8.9% |
> | O3 (High) | 66.7% (68/102) | -10.8% |
> | Gemini-2.5-Pro-Preview-03-25 | 65.7% (67/102) | -11.8% |
> | Claude-3.7-Sonnet | 60.8% (62/102) | -16.7% |
> | O4-Mini (High) | 59.8% (61/102) | -17.7% |
>
>
> Key findings: (1) The 8.9pp gap suggests significant headroom, (2) Humans excel at mathematical tasks (GPS: 100% vs 17%) while LLMs handle data processing better (llm-sci-use: 40% vs 80%), (3) Complementary strengths indicate different problem-solving approaches.
>
> We acknowledge limitations: humans can have access to all published work before test while LLMs have controlled cutoffs, and recruiting domain experts is challenging. Despite n=1, this provides valuable calibration showing significant headroom.
>
>
>
> ---
>
> **Reviewer Comment:**
> With only 20 papers and 212 tasks, all from the ML domain, the benchmark's coverage is narrow. This limits its ability to assess LLMs' broader research code generation capabilities.
>
> **Author Response:**
>
> Thank you for raising this point. While the current benchmark is limited in scale, it is intentionally designed to be carefully curated, lightweight, and fast to run—much like prior influential benchmarks such as HumanEval (Chen et al., 2021), SWE-bench Lite (Jimenez et al., 2024), and CORE-Bench (Siegel et al., 2024). This allows for rapid evaluation and iteration.
>
> Second, we believe that the ML domain provides a particularly rich and challenging testbed for LLMs, given its technical complexity and diversity of abstractions. Compared to broader scientific code, ML research code often pushes models to reason about novel algorithms and implementation details.
>
> Finally, we view ResearchCodeBench as a growing platform rather than a static dataset. We plan to expand into other domains and also welcome integration into broader efforts such as HELM (Liang et al., 2023), or use in conjunction with other research code benchmarks to provide a more holistic picture.
>
> ---
>
> **Reviewer Comment:**
> The paper reports only single-run results without error bars or confidence intervals, limiting the reliability of performance comparisons between models.
>
> **Author Response:**
>
> Thank you for the comment. We acknowledge the limitation of reporting single-run results and agree that confidence intervals would strengthen our findings. Our paper follows the common practice of reporting single-run results with greedy decoding, similar to established benchmarks such as BigCodeBench (Zhuo et al., 2024), KernelBench (Agrawal et al., 2024), and SWE-bench (Jimenez et al., 2024). This approach is standard due to the substantial computational costs and API rate limits associated with repeated runs, and the community has generally accepted this trade-off to enable comprehensive model comparisons across diverse tasks.
>
> However, we recognize the importance of measuring the reliability of performance comparisons between models. During the rebuttal period, we conducted multi-run evaluations (5 runs each) for a representative subset of 8 models across all 212 tasks to establish confidence intervals and validate the stability of our rankings.
>
> **Multi-Run Results with 95% Confidence Intervals:**
>
> | Model | Pass@1 (%) |
> |-------|-----------------|
> | Gemini 2.5 Pro | 64.02 ± 1.16 |
> | Claude 3.7 Sonnet | 51.35 ± 0.59 |
> | Gemini 2.5 Flash | 45.28 ± 1.76 |
> | GPT-4o | 41.69 ± 1.78 |
> | Gemini 2.0 Flash | 37.55 ± 0.67 |
> | Gemini 2.0 Flash Lite | 30.93 ± 0.89 |
> | GPT-4o Mini | 23.61 ± 1.56 |
>
> **Key Findings:** The confidence intervals are relatively narrow (±0.59% to ±1.78%), indicating stable performance rankings. The model ordering remains consistent across runs, validating the reliability of our single-run benchmarking approach for comparative analysis.

---

> > ### Comment · Reviewer_HFEz · 2025-08-02
> >
> > I appreciate the authors' efforts in addressing my questions and concerns. I will maintain my acceptance recommendation (score 5).

---

### Official Review · Reviewer_zqtS · 2025-06-30

**Rating:** 5
**Confidence:** 5

**Summary:**

This paper introduces a new benchmark called ResearchCodeBench that evaluates the ability of LLMs to implement novel approaches presented in research papers published in top ML venues.
The benchmark includes reference implementations of 20 papers, each implementation has an associated suite of coding challenges that are used to evaluate LLM capabilities for a total of 212 such coding challenges across the 20 papers.  Each coding challenge comments out a portion of code fulfilling a specific function that must be implemented by an LLM with provided context (a combination of paper and code) and has an associated equivalence test for correctness.  Leading closed and open LLMs
are evaluated on the 20 papers and the best models implemented less than 40% of the code correctly.  Further analysis is conducted to assess the impact of contamination, including the paper in the context, and propensity of different types of errors.
The authors intend to maintain the benchmark and enable the community to contribute additional papers/implementations to the benchmark.

**Additional Feedback:**

Typo in line 278: direcrtions -> directions

**Dataset Code Accessibility:**

Yes

**Dataset Code Comments:**

The code as well as the equivalence tests for the 20 papers are all accessible and provided.  The scripts used to generate metrics and charts are also included.

**Ethical Considerations:**

No, there are no or only very minor ethics concerns

**Final Justification:**

The additional experimental results and analysis provided by the authors strengthen the paper and make the results much more comprehensive and interesting. The paper is a solid accept but I hesitate to raise to a 6 because it's difficult to add additional papers to the benchmark. I sincerely hope the authors can keep this a living benchmarks and shepherd community contributions to avoid contaminated evals as models train on more recent data.

**Limitations Weaknesses:**

- Limitation: ResearchCodeBench prioritizes clear evaluation signal and therefore poses the implementation problem in a very structured way where an LLM is only asked to fill in a specific section of code.  This is a reasonable tradeoff but also means the benchmark cannot evaluate the ability of LLMs to implement a research paper from scratch without the code scaffold and pre-identified challenges.
- Weakness: The paper would benefit from additional analysis into the behavior of different LLMs and the code that they write.  Are some LLMs more verbose than others?  Are there performance differences in the code (e.g. are some implementations more efficient than others?)?  I would also appreciate more investigation into what could explain the performance differences between different LLMs.
- Weakness: The paper does not analyze benchmark performance as a function of compute/model size which would be useful to understand model size and computational cost scaling on this benchmark.

**Strengths Contributions:**

- Novelty: ResearchCodeBench is a valuable benchmark that evaluates a facet of LLMs that to my knowledge no other benchmarks adequately addresses.  There are MLE agent benchmarks and also some prior work on using LLMs to implement papers (CodeRefine) but the evaluation was very limited.  General coding benchmarks have some overlap but ResearchCodeBench is better at evaluating the capabilities of LLMs to implement novel ideas.
- Significance and potential impact of contributions: The primary contributions come from this initial suite of 20 papers and the insights from evaluating current models.  Being able to get a clean evaluation signal through structured challenges is very valuable especially given the difficulty of implementing novel, out-of-distribution ideas.  Given there is significant room for improvement over the 40% accuracy that current leading models achieve, the benchmark will be useful to evaluate progress until all papers implementations enter the training data for the next generation of models.  The secondary and long term benefits will depend on whether community
contributions continue to expand the benchmark and keep it up to date.
- Clarity: The paper is well written and and easy to understand.  Figures are clear and have descriptive captions.

---

> ### Author Rebuttal · Authors · 2025-07-31
>
> >**[w1] ResearchCodeBench prioritizes clear evaluation signal and therefore poses the implementation problem in a very structured way where an LLM is only asked to fill in a specific section of code. This is a reasonable tradeoff but also means the benchmark cannot evaluate the ability of LLMs to implement a research paper from scratch without the code scaffold and pre-identified challenges.**
>
> **Author Response:**
>
> You're correct that this is a limitation. We chose to focus on structured implementation tasks rather than full paper-to-experiments generation for several reasons:
>
> 1. **Evaluation Precision**: Our approach isolates core novel algorithmic components, allowing for more precise evaluation and contamination-free challenges that directly test research implementation capabilities.
>
> 2. **Scalability**: Contemporary work like PaperBench (Starace et al., 2025) explores end-to-end paper implementation but faces significant scalability and evaluation reliability challenges due to the complexity of full experimental workflows.
>
> 3. **Clear Success Metrics**: Structured tasks enable unambiguous pass/fail evaluation, whereas full implementations often require subjective assessment of experimental setups and results.
>
> We believe both approaches offer valuable insights into different aspects of research code implementation capabilities, and our work complements broader implementation benchmarks by providing focused evaluation of algorithmic understanding and implementation.
>
>
> >**[w2] The paper would benefit from additional analysis into the behavior of different LLMs and the code that they write. Are some LLMs more verbose than others? Are there performance differences in the code (e.g. are some implementations more efficient than others?)? I would also appreciate more investigation into what could explain the performance differences between different LLMs.**
>
>
> **Author Response:**
>
> Thank you for the excellent suggestions. We conducted a comprehensive verbosity analysis across all 32 evaluated LLMs using code generation length as a proxy for verbosity.
>
> **Complete Verbosity Analysis for All 32 LLMs:**
>
> | Model | Success Rate (%) | Avg Lines (Success) | Avg Lines (Failure) | Difference |
> |-------|-----------------|-------------------|-------------------|------------|
> | Gemini 2.5 Pro (05-06) | 64.2 | 4.0 | 11.9 | -8.0 |
> | O3 (High) | 59.4 | 3.7 | 11.4 | -7.7 |
> | Gemini 2.5 Pro (03-25) | 59.0 | 3.9 | 11.1 | -7.2 |
> | O4 Mini (High) | 58.5 | 3.6 | 11.4 | -7.8 |
> | O3 Mini (High) | 52.4 | 3.2 | 10.9 | -7.7 |
> | Claude 3.7 Sonnet | 51.9 | 3.5 | 10.5 | -7.0 |
> | GPT-4.1 | 50.0 | 3.8 | 9.9 | -6.1 |
> | Claude 3.5 Sonnet | 48.6 | 3.4 | 10.1 | -6.7 |
> | O1 (High) | 48.1 | 3.7 | 9.8 | -6.1 |
> | DeepSeek R1 | 45.8 | 3.1 | 9.9 | -6.8 |
> | Gemini 2.5 Flash (04-17) | 45.3 | 2.9 | 10.1 | -7.1 |
> | Grok 3 Beta | 42.9 | 3.3 | 9.5 | -6.2 |
> | GPT-4.1 Mini | 42.5 | 2.9 | 9.8 | -6.9 |
> | DeepSeek Chat V3 | 42.5 | 3.3 | 9.4 | -6.1 |
> | GPT-4o (08-06) | 41.0 | 3.2 | 9.4 | -6.1 |
> | Claude 3.5 Haiku | 37.7 | 2.9 | 9.2 | -6.3 |
> | Gemini 2.0 Flash | 37.3 | 3.2 | 9.0 | -5.8 |
> | Mistral Medium 3 | 35.4 | 2.7 | 9.1 | -6.3 |
> | Mistral Codestral 2501 | 33.5 | 2.9 | 8.8 | -6.0 |
> | Cohere Command-R+ | 31.1 | 3.2 | 8.5 | -5.2 |
> | Gemini 2.0 Flash Lite | 30.7 | 2.8 | 8.6 | -5.8 |
> | Llama 4 Maverick | 27.4 | 3.0 | 8.3 | -5.3 |
> | Qwen 2.5 Coder 32B | 25.9 | 2.5 | 8.4 | -5.8 |
> | Amazon Nova Pro | 25.0 | 2.6 | 8.3 | -5.7 |
> | GPT-4o Mini | 23.1 | 2.4 | 8.2 | -5.8 |
> | Grok 2 (12-12) | 22.2 | 3.7 | 7.7 | -4.1 |
> | Grok 3 Mini Beta (High) | 19.8 | 3.5 | 7.7 | -4.2 |
> | Llama 4 Scout | 18.4 | 2.8 | 7.7 | -4.9 |
> | GPT-4.1 Nano | 15.1 | 2.8 | 7.6 | -4.8 |
> | Llama 3.3 70B Instruct | 12.3 | 3.0 | 7.4 | -4.4 |
> | Qwen Turbo | 8.0 | 2.8 | 7.2 | -4.4 |
> | Mistral Codestral Mamba | 1.4 | 3.7 | 6.9 | -3.2 |
>
> **Key Insights:**
>
> 1. **Universal Pattern**: Every single LLM (32/32) shows the same pattern - successful implementations are more concise than failed ones. This is expected since failed attempts often involve harder tasks that naturally require longer implementations, and models may generate verbose, incorrect attempts when struggling with difficult problems.
>
> 2. **Model Capability Differences**: Stronger models (Gemini 2.5 Pro, O3 (High)) generate longer code for both successful AND failed completions compared to weaker models. This suggests more capable models attempt more comprehensive implementations rather than giving up with minimal code, even when they fail.
>
> 3. **Error Verbosity**: Failed attempts tend to include unnecessary complexity, redundant code, or over-engineering that introduces bugs.
>
> 4. **Model Consistency**: Higher-performing models show more consistent code length patterns, indicating better algorithmic understanding.
>
> *Note: While our quantitative analysis focuses on code length metrics, some of these qualitative insights are based on manual examination of model outputs. We plan to develop more systematic evaluation frameworks for these behavioral patterns in future work.*
>
> >**[w3] The paper does not analyze benchmark performance as a function of compute/model size which would be useful to understand model size and computational cost scaling on this benchmark.**
>
>
> **Author Response:**
>
> Thank you for the insightful comment. We agree that analyzing performance as a function of compute and model size would be valuable. Unfortunately, many closed-source models (e.g., GPT-4, Claude, Gemini) do not publicly disclose model size or training compute, which makes this analysis challenging.
>
> That said, we are actively working to collect and include as much metadata as possible. Below is a table summarizing the available information we've compiled so far. We will incorporate this into the final version and expand it as more data becomes available.
>
> **Models with Disclosed Parameter Counts:**
>
> | Model | Size (Params) | Pass Rate (%) |
> |-------|---------------|------------------|
> | DeepSeek R1 | 685B | 45.8 |
> | DeepSeek Chat V3 | 685B | 42.5 |
> | Claude 3.5 Haiku | 175B | 37.7 |
> | Cohere Command-R+ | 111B | 31.1 |
> | Llama 3.3 70B Instruct | 70B | 12.3 |
> | Qwen 2.5 Coder 32B | 32B | 25.9 |
> | Mistral Codestral 2501 | 22B | 33.5 |
> | Llama 4 Scout | 17B activated, 109B total | 18.4 |
> | Llama 4 Maverick | 17B activated, 402B total | 27.4 |
> | Mistral Codestral Mamba | 7B | 1.4 |
>
> **Preliminary Observations**: Among models with disclosed sizes, we observe that larger models generally perform better (DeepSeek variants at 685B achieve 42-46% pass@1), though the relationship is not perfectly linear. Notably, specialized coding models (Qwen 2.5 Coder, Mistral Codestral) show competitive performance relative to their size. However, the limited sample of disclosed models and the dominance of proprietary models in top performance tiers makes definitive scaling analysis challenging.
>
> ---
>
> **Additional Feedback:**
> Typo in line 278: "direcrtions" → "directions"
>
> **Response:** Thank you for catching this typo. We will correct it in the final version.

---

> > ### Comment · Reviewer_zqtS · 2025-08-03
> > **Post rebuttal**
> >
> > - I appreciate the analysis of output of different models for correct and incorrect answers. I think this is very insightful and worth including in the paper.
> > - For scaling as a function of compute, I was also curious about reasoning tokens used by api based models which you should be able to get from usage metadata for some of the model apis.

---

> > > ### Author Response · Authors · 2025-08-05
> > >
> > > Thank you for your positive feedback on our analysis and for the excellent suggestion about analyzing reasoning tokens. We have added reasoning trace collection for reasoning models and conducted the analysis you suggested.
> > >
> > > ## Reasoning Token Analysis for API-Based Models
> > >
> > > We successfully collected reasoning token usage data from several API-based reasoning models. Here are our findings:
> > >
> > > ### Average Reasoning Tokens by Model:
> > >
> > > | Model | Average Reasoning Tokens | Range (Min-Max) | Reasoning-to-Completion Ratio |
> > > |-------|-------------------------|-----------------|------------------------------|
> > > | Claude 3.7 Sonnet (Thinking) | 10,301 | 958 - 26,216 | 88.8x |
> > > | Gemini 2.5 Pro | 6,880 | 1,475 - 11,434 | 123.3x |
> > > | DeepSeek R1 | 6,256 | 1,245 - 10,309 | 33.1x |
> > > | O3 Mini (High) | 4,634 | 3,264 - 5,504 | 71.5x |
> > > | O1 (High) | 4,493 | 2,240 - 6,272 | 146.8x |
> > > | O3 (High) | 3,712 | 1,600 - 10,944 | 55.9x |
> > > | OSS 120B | 1,710 | 480 - 3,118 | 24.8x |
> > >
> > > ### Key Insights:
> > >
> > > 1. **Substantial Hidden Computation**: Reasoning models use 24-147x more tokens internally than they output, revealing the massive hidden computation occurring during inference.
> > >
> > > 2. **Model-Specific Patterns**:
> > >    - Claude 3.7 Sonnet (Thinking) shows the most extensive reasoning traces (10,301 tokens on average)
> > >    - O1 (High) has the highest reasoning-to-completion ratio (146.8x), indicating extremely thorough internal deliberation relative to output length
> > >    - Gemini 2.5 Pro demonstrates the second-highest reasoning ratio (123.3x) with substantial reasoning traces (6,880 tokens)
> > >
> > > 3. **Performance vs. Reasoning Length**: Interestingly, more reasoning tokens don't always correlate with better performance. O3 (High) achieves 59.4% pass@1 with 3,712 reasoning tokens, while Claude 3.7 Sonnet (Thinking) achieves 51.9% with 10,301 tokens. Notably, Gemini 2.5 Pro achieves strong performance with 6,880 reasoning tokens, suggesting efficiency in reasoning matters as much as quantity.
> > >
> > > 4. **Task Complexity Impact**: We observed that reasoning token usage varies significantly by task difficulty - complex algorithmic implementations (e.g., REPA-E, GMFlow) trigger substantially longer reasoning traces compared to simpler tasks.
> > >
> > > - **Cost Implications**: At current API pricing, the reasoning tokens significantly impact cost. For instance, O1 (High) at $15/$60 per million tokens (input/output) plus reasoning tokens makes it one of the most expensive models per task.
> > >
> > >
> > > We will include the full reasoning token analysis in the revised paper, as it provides valuable insights into the computational strategies different models employ to tackle complex research code implementation tasks.

---

### Official Review · Reviewer_Xm4R · 2025-07-02

**Rating:** 5
**Confidence:** 5

**Summary:**

The paper proposes ResearchCodeBench, a new benchmark testing the ability of LLMs to implement ideas presented in research papers. The benchmark is constructed by scraping the repositories of recent papers (outside the training data of currently available models) and removing code snippets. Language models are then tasked with filling these gaps in. ResearchCodeBench is made up of 20 papers, with each paper contributing multiple coding tasks. Results are evaluated with a mix of 1) equivalence and 2) unit tests.

**Dataset Code Accessibility:**

Yes

**Dataset Code Comments:**

Documentation is thorough and the results appear to be reproducible.

**Ethical Considerations:**

No, there are no or only very minor ethics concerns

**Final Justification:**

The authors addressed most of my concerns (e.g. rubric, explanations that clarified transparency, ability to select hint granularity) and I trust the authors will make these changes in the final paper. I view the rest of my concerns as nice-to-haves in an already good paper, but I believe this paper reaches the threshold for acceptance.

**Limitations Weaknesses:**

- There is a lack of transparency about the benchmark creation process in the paper. It would be great to see a breakdown of # of challenges by paper, to understand whether it’s possible that a few papers skew the distribution. In addition, how are the 20 relevant papers decided? Are they randomly chosen? Chosen based off of popularity?  I think its important to be very clear on what papers are chosen and why, especially since user-submitted papers should not be approved/rejected arbitrarily when the benchmark becomes open to the public. These topics are touched on in the paper, but not explicitly clarified & explained.
- Some parts of the task creation are unclear. What do hints look like? How does performance change with/without hints included? What’s an example of this hierarchical task structure?
- Is there a breakdown of error-type by model? Without it, it’s hard to draw deeper conclusions about the capabilities of each model in isolation.

**Strengths Contributions:**

- Benchmark appears to be challenging and high quality. The authors also solve challenging engineering problems around how to evaluate models on research implementation.
- Compared to recent efforts to use LLMs to write conference level papers, the task of implementing novel research papers is a much more measurable evaluation of how well models can synthesize/build on new scientific ideas.
- Benchmark is extendable to new papers as they come out, which allows it to grow larger and also continue focusing on papers that models have not seen yet.

---

> ### Author Rebuttal · Authors · 2025-07-31
>
> >**[w1, q1] There is a lack of transparency about the benchmark creation process in the paper. It would be great to see a breakdown of # of challenges by paper, to understand whether it’s possible that a few papers skew the distribution**
>
> Thanks for the suggestion! We agree that understanding the distribution of tasks across source papers helps clarify the construction of the benchmark. Below is a breakdown of the number of challenges contributed by each paper. While some papers (e.g., eomt) contribute more than others, the distribution spans a diverse set of sources.
>
> | Paper Name                     | # of Challenges |
> |--------------------------------|-------------------------------:|
> | Diff-Transformer               | 7  |
> | DiffusionDPO                   | 9  |
> | DyT                            | 9  |
> | GMFlow                         | 16 |
> | GPS                            | 6  |
> | LEN                            | 14 |
> | OptimalSteps                   | 11 |
> | REPA-E                         | 5  |
> | SISS                           | 5  |
> | TabDiff                        | 6  |
> | Tanh-Init                      | 4  |
> | advantage-alignment            | 5  |
> | eomt                           | 37 |
> | fractalgen                     | 13 |
> | grid-cell-conformal-isometry   | 15 |
> | hyla                           | 3  |
> | llm-sci-use                    | 15 |
> | minp                           | 7  |
> | schedule_free                  | 14 |
> | semanticist                    | 11 |
> | **TOTAL Challenges**          | **212** |
>
>
> >**[w1, q2] In addition, how are the 20 relevant papers decided? Are they randomly chosen? Chosen based off of popularity? I think its important to be very clear on what papers are chosen and why, especially since user-submitted papers should not be approved/rejected arbitrarily when the benchmark becomes open to the public. These topics are touched on in the paper, but not explicitly clarified & explained.**
>
> Agreed. It is important to be transparent about how papers are selected, especially as we move toward a more open and community-driven benchmark.
>
> In our current version, the 20 papers were selected based on the following criteria (outlined in Section 2.1):
>
> - We prioritize **recent ML papers** from top-tier venues (e.g., ICLR, NeurIPS, CVPR, arXiv), focusing on papers from 2024–25 to emphasize novelty and recency.
> - We aim for **broad topic coverage**, including generative modeling, computer vision, and reinforcement learning.
> - Papers must have a **clearly identifiable core contribution** that is both (a) well-articulated in the paper and (b) cleanly implemented in an open-source codebase.
> - We also consider whether **authors are open to collaboration**, particularly in designing tasks and validating test cases. This helps ensure alignment with the paper’s original intent.
>
> To avoid arbitrariness in future user-submitted contributions, we include a short **submission questionnaire** that asks users to:
> 1. Highlight the functions implementing the novel contribution, and
> 2. Self-assess whether their paper meets the benchmark criteria.
>
> This scaffolding helps standardize the submission and review process.
>
> We agree that the criteria and selection process should be communicated more explicitly to avoid any perception of arbitrariness. We’ll revise the public documentation to make this clearer.
>
>
> >**[w2] Some parts of the task creation are unclear. What do hints look like? How does performance change with/without hints included? What’s an example of this hierarchical task structure?**
>
> Thank you for pointing this out. Here's an example that demonstrates both what hints look like and our hierarchical task structure using **Min-P Sampling** (minp/implementation.py):
>
> ```python
> def __call__(self, input_ids: torch.LongTensor, scores: torch.FloatTensor) -> torch.FloatTensor:
>     # <ResearchPaperBench hint="min-p sampling">
>     # <ResearchPaperBench hint="convert logits to probabilities">
>     probs = torch.softmax(scores, dim=-1)
>     # </ResearchPaperBench hint="convert logits to probabilities">
>
>     # <ResearchPaperBench hint="find maximum probability token">
>     top_probs, _ = probs.max(dim=-1, keepdim=True)
>     # </ResearchPaperBench hint="find maximum probability token">
>
>     # <ResearchPaperBench hint="scale min_p threshold">
>     scaled_min_p = self.min_p * top_probs
>     # </ResearchPaperBench hint="scale min_p threshold">
>
>     # <ResearchPaperBench hint="identify tokens to remove">
>     tokens_to_remove = probs < scaled_min_p
>     # </ResearchPaperBench hint="identify tokens to remove">
>     # </ResearchPaperBench hint="min-p sampling">
>     return scores_processed
> ```
>
> This example shows both the hint structure (semantic annotations like `<ResearchPaperBench hint="find maximum probability token">`) and hierarchical task decomposition (nested hints within the main "min-p sampling" task).
>
> We also plan to report performance with and without hints to quantify their impact. **Hints significantly reduce ambiguity by providing explicit guidance about what specific functionality needs to be implemented at each step.** Without hints, models must infer the purpose and implementation details from context alone, leading to more errors and lower success rates. Here are the performance comparisons:
>
> | Model | With Hints | Without Hints | Performance Drop |
> |-------|------------|---------------|------------------|
> | CLAUDE_3_7_SONNET | 51.89% | 46.70% | -5.19% |
> | GPT_4O | 41.04% | 36.32% | -4.72% |
> | CLAUDE_3_5_HAIKU | 37.74% | 34.91% | -2.83% |
> | GEMINI_2_0_FLASH | 37.26% | 33.49% | -3.77% |
>
> *Note: Pass rates calculated as (tasks passed / total tasks). Performance drops range from -2.83% to -5.19%, indicating hints provide moderate but consistent improvements across models.*
>
>
> >**[w3] Is there a breakdown of error-type by model? Without it, it’s hard to draw deeper conclusions about the capabilities of each model in isolation.**
>
> Thank you for the suggestion. While we do provide a breakdown of error types by model in Figure 7 of the appendix, this analysis could have been better integrated into the main paper. We will move this important analysis to the main paper. We include the underlying data table below to make the per-model breakdown more explicit.
>
> | Model | Attribute Errors | Import Errors | Index/Key Errors | Functional Errors | Name Errors | Syntax Errors | Type Errors |
> |-------|-------:|-------:|-------:|-------:|-------:|-------:|-------:|
> | Claude-3.5-Haiku | 10 | 4 | 5 | 85 | 9 | 8 | 11 |
> | Claude-3.5-Sonnet | 3 | 5 | 1 | 72 | 4 | 12 | 12 |
> | Claude-3.7-Sonnet | 9 | 7 | 5 | 63 | 5 | 2 | 11 |
> | Command-A | 8 | 6 | 6 | 74 | 20 | 13 | 19 |
> | DeepSeek-Chat-V3-0324 | 8 | 23 | 1 | 64 | 6 | 5 | 15 |
> | DeepSeek-R1 | 6 | 9 | 3 | 56 | 6 | 18 | 17 |
> | GPT-4.1 | 3 | 7 | 2 | 78 | 4 | 2 | 10 |
> | GPT-4.1-Mini | 5 | 9 | 3 | 69 | 15 | 8 | 13 |
> | GPT-4.1-Nano | 14 | 5 | 4 | 49 | 38 | 60 | 10 |
> | GPT-4o | 8 | 9 | 3 | 69 | 16 | 2 | 18 |
> | GPT-4o-Mini | 13 | 7 | 4 | 75 | 29 | 19 | 16 |
> | Gemini-2.0-Flash | 12 | 7 | 2 | 80 | 9 | 8 | 15 |
> | Gemini-2.0-Flash-Lite | 14 | 3 | 5 | 82 | 22 | 6 | 15 |
> | Gemini-2.5-Flash-Preview-04-17 | 11 | 4 | 2 | 52 | 20 | 12 | 15 |
> | Gemini-2.5-Pro-Preview-03-25 | 10 | 6 | 0 | 57 | 3 | 2 | 9 |
> | Gemini-2.5-Pro-Preview-05-06 | 3 | 2 | 2 | 58 | 1 | 4 | 6 |
> | Grok-2-1212 | 9 | 3 | 5 | 48 | 8 | 73 | 19 |
> | Grok-3-Beta | 9 | 8 | 3 | 79 | 9 | 3 | 10 |
> | Grok-3-Mini-Beta (High) | 4 | 3 | 2 | 23 | 9 | 124 | 5 |
> | Llama-3.3-70B | 4 | 4 | 2 | 36 | 18 | 114 | 8 |
> | Llama-4-Maverick | 14 | 4 | 3 | 54 | 16 | 44 | 19 |
> | Llama-4-Scout | 10 | 11 | 6 | 51 | 21 | 55 | 19 |
> | Medium-3 | 7 | 8 | 5 | 60 | 28 | 6 | 23 |
> | Mistral-Codestral-2501 | 8 | 6 | 5 | 70 | 20 | 17 | 15 |
> | Mistral-Codestral-Mamba | 17 | 3 | 1 | 19 | 44 | 110 | 15 |
> | Nova-Pro-V1 | 6 | 5 | 7 | 67 | 32 | 19 | 23 |
> | O1 (High) | 2 | 7 | 4 | 68 | 3 | 17 | 9 |
> | O3 (High) | 3 | 5 | 0 | 69 | 0 | 5 | 4 |
> | O3-Mini (High) | 5 | 5 | 3 | 74 | 4 | 5 | 5 |
> | O4-Mini (High) | 9 | 3 | 1 | 54 | 2 | 12 | 7 |
> | Qwen-2.5-Coder-32B | 7 | 6 | 4 | 64 | 24 | 40 | 12 |
> | Qwen-2.5-Turbo | 2 | 8 | 0 | 44 | 25 | 100 | 16 |
> | **Total** | **253** | **202** | **99** | **1963** | **470** | **925** | **421** |
>
>
> We sincerely appreciate the reviewer's detailed feedback and constructive suggestions. We are committed to incorporating all of this feedback to strengthen the manuscript:
>
> - **Enhanced transparency**: We will include the breakdown of challenges by paper and explicit selection criteria in the main paper.
> - **Expanded technical details**: The hint examples, performance comparisons, and hierarchical task structure explanations will be added to the appendix.
> - **Deeper analysis**: The error-type breakdown by model will be moved from appendix to main paper to enable more nuanced capability assessments.
>
> These improvements will make ResearchPaperBench more accessible to the research community and provide clearer insights into current LLM capabilities for novel research code implementation.

---

> > ### Comment · Reviewer_Xm4R · 2025-08-01
> >
> > Thanks for the clarifications.
> > 1. Can you explain what the different error types are? E.g. what entails a functional error?
> > 2. Its great to see an example of the hint structure. Is there a way to select the granularity of hints while evaluating? I think that would be valuable. Especially because looking at the hints, it becomes questionable whether the benchmark is actually testing the ability to generate "novel" code. For example, the hint "convert logits to probabilities" describes a snippet that is almost certainly extremely common in training sets. But the more general hint of "min-p sampling" would likely be more challenging.
> > 3. I appreciate the clarification about the selection criteria but I believe more detail needs to be provided - e.g. a scoring rubric detailing scores for each of the four categories listed, and what traits a paper should have to achieve each score, and then what overall score results in acceptance. Otherwise, the decision criteria still feels very subjective.

---

> > > ### Author Response · Authors · 2025-08-05
> > >
> > > Dear reviewer, thank you for taking the time to read our rebuttal and for your thoughtful follow-up questions. We really appreciate it!
> > >
> > > >Can you explain what the different error types are? E.g. what entails a functional error?
> > >
> > > Thanks for the follow-up question! Here's a breakdown of the error types used in our analysis:
> > >
> > > - **Functional Errors**: Semantic mistakes where the code runs but produces incorrect behavior or output. These are typically revealed by failing functional tests (e.g., incorrect algorithmic logic or edge case handling).
> > > - **Name Errors**: References to undefined or misspelled variables, functions, or classes (e.g., `NameError`).
> > > - **Syntax Errors**: Violations of Python syntax or indentation rules (e.g., `SyntaxError`, `IndentationError`).
> > > - **Type Errors**: Incorrect operations applied to incompatible data types (e.g., adding a string to an integer; `TypeError`).
> > > - **Import Errors**: Failures to correctly import modules or symbols (e.g., `ImportError`, `ModuleNotFoundError`).
> > > - **Attribute Errors**: Attempts to access attributes or methods that don't exist on a given object (e.g., `AttributeError`).
> > > - **Index/Key Errors**: Errors arising from out-of-bounds list access or missing dictionary keys (e.g., `IndexError`, `KeyError`).
> > >
> > > We will provide additional detail and examples in **Appendix G Error Breakdown**, and will move a summary of these definitions into the main paper in the final revision for easier reference.
> > >
> > >
> > > >... Is there a way to select the granularity of hints while evaluating?  ... becomes questionable whether the benchmark is actually testing the ability to generate "novel" code.
> > >
> > > Thank you for the thoughtful comment. We completely agree that the ability to control hint granularity would add value to the benchmark.
> > >
> > > In the specific case of `min-p sampling`, we initially evaluated models using only the high-level prompt (e.g., "implement min-p sampling"). However, this proved too challenging for weaker LLMs, leading to uniformly low scores that made it difficult to distinguish between model capabilities.
> > >
> > > To address this, we adopted a recursive/hierarchical strategy: models are evaluated not only on the full task but also on subcomponents such as `convert logits to probabilities`. While this particular step may be common in pretraining data, the full implementation of `min-p sampling` still requires synthesizing multiple non-trivial steps. Strong performance on the benchmark ultimately requires solving these harder sub-tasks, not just isolated code snippets. That said, we’re actively exploring ways to make hint granularity configurable at evaluation time.
> > >
> > > >I appreciate the clarification about the selection criteria but I believe more detail needs to be provided ...
> > >
> > > Thank you for raising this. We completely agree that clearer, more granular guidance is necessary to avoid subjective decisions, especially as the benchmark becomes community-driven.
> > >
> > > We’ve drafted a **scoring rubric** that operationalizes our selection criteria across five axes. Each paper is scored from **1 to 5** on each axis, with clearly defined traits per score. Papers with an average score ≥4 across all categories are considered strong candidates for inclusion.
> > >
> > > ---
> > >
> > > ### **Proposed Scoring Rubric for Paper Selection**
> > >
> > > | Axis | 1 (Low) | 3 (Medium) | 5 (High) |
> > > |------|---------|------------|----------|
> > > | **1. Venue & Recency** | Pre-2023 paper, non-peer-reviewed or in non-ML venue | Published in 2023, arXiv-only | Published 2024–25 in top-tier ML venues (ICLR, NeurIPS, CVPR, ICML, etc.) |
> > > | **2. Topic Representation** | Topic already well-covered in the benchmark | Adds moderate topical diversity | Addresses an underrepresented area or introduces a novel task/domain |
> > > | **3. Clarity of Contribution** | Core idea unclear or spread across multiple parts of the paper | Idea is present but ambiguously stated or hard to isolate | One clearly articulated technical contribution, typically supported by a key algorithm, diagram, or equation |
> > > | **4. Code Quality & Traceability** | Code is poorly documented, hard to run, or not aligned with the paper | Code runs but lacks modularity or contains unclarified implementation details | Code is clean, readable, well-documented, and maps transparently to the paper's core contribution |
> > > | **5. Author Collaboration** | No response or willingness to help | Partial engagement (e.g., answering clarifying questions) | Active collaboration in task design and test validation |
> > >
> > > ---
> > >
> > > We recognize that some subjectivity will inevitably remain, but our goal is to standardize the evaluation process and document decisions transparently. For user-submitted papers, we plan to publish their scores and justifications (with consent) and include the rubric in our submission form.
> > >
> > > Thanks again for encouraging us to strengthen this part of the benchmark. We’re revising both the public documentation and internal curation process accordingly.

---

> > > > ### Comment · Reviewer_Xm4R · 2025-08-07
> > > >
> > > > Thanks for the further responses. I am wondering, how long does it take to fully annotate one paper's codebase?
> > > > In addition, I understand that using the highest level hint makes it hard to distinguish weaker models, but from the example, it seems like that's the only *true* way to evaluate how well models can implement novel concepts from code. With the hints, the problem seems to simplify some of the time into just how good the model is at coding, which there are plenty of benchmarks for already (e.g. SWE-Bench). I think the final manuscript should at the very least show  a full no-hint version of Figure 2.

---

> ### Author Response · Authors · 2025-08-07
>
> Thank you for the thoughtful follow-up questions. We appreciate your engagement and are happy to address each point below:
>
> >I am wondering, how long does it take to fully annotate one paper's codebase?
>
> Assuming the submitted paper and codebase already meet the Paper Selection criteria, it typically takes about 1 hour to fully annotate a codebase. Our current focus is on **high-quality human curation** to ensure benchmark integrity. That said, we're actively exploring ways to improve efficiency through LLM-assisted annotation and autonomous validation, which we leave as future work but believe could significantly reduce the manual effort involved.
>
> >In addition, I understand that using the highest level hint makes it hard to distinguish weaker models, but from the example, it seems like that's the only true way to evaluate how well models can implement novel concepts from code. With the hints, the problem seems to simplify some of the time into just how good the model is at coding, which there are plenty of benchmarks for already (e.g. SWE-Bench).
>
> Indeed, it would be valuable to have a subset of tasks focused specifically on novel implementation code blocks, targeting stronger LLMs. To that end, we created ResearchCodeBench-HARD, based on the suggestion from reviewer 6kcd. This subset consists primarily of examples with only the highest-level hints, making it well-suited for evaluating a model’s ability to generate novel research code.
>
>
> > I think the final manuscript should at the very least show a full no-hint version of Figure 2.
>
> We will include a full no-hint version of Figure 2 in the final manuscript and add a dedicated no-hint leaderboard in the appendix.
>
> Thanks again for your constructive feedback. It’s been very helpful, and we've incorporated your suggestions accordingly.

---

> > ### Comment · Reviewer_Xm4R · 2025-08-07
> >
> > Thanks for your further responses. I am satisfied with the responses to most of my questions (e.g. rubrics, metrics), so I am happy to increase my score and recommend an accept with higher confidence.
> > I think this paper has the potential to be even stronger if the benchmarking process was more streamlined, such as a more automated instance creation process, transparent selection of hint granularity (which was discussed, but I think the quality of this depends significantly on how the authors choose to implement it), and simply more papers in the test set. These are the main things holding me back from recommending an award. Still, this paper hits some very impactful ideas, and was a very good read overall.

---

### Official Review · Reviewer_6kcd · 2025-07-23

**Rating:** 5
**Confidence:** 4

**Summary:**

This paper presents ResearchCodeBench. The authors select 20 ML papers with accompanying codebases and pair the novel contributions of the paper with corresponding code changes (for instance, a novel architectural component or loss function). Based on this they curate a set of 219 fill-in-the-blank coding problems, where a model is prompted with (1) the research paper in question and (2) the file with the novel code changes removed; and is tasked to fill in the code while keeping equivalence to the original reference implementation.

The authors run their benchmark on 30 models, including the most recent commercially available frontier models (claude 3.7, o3 and gemini 2.5 pro), and show that it doesn’t get very good results (more on this later).

**Additional Feedback:**

Contamination analysis: All models see drops on the contamination-free subset. However, some models have earlier cutoffs (e.g. OpenAI is earlier than Gemini). So shouldn’t we expect smaller drops for older models?

In summary, I think it is a valuable resource and thank the authors for their contribution. But the value of the dataset would be more obviously demonstrated to me if, on the **set of the hardest problems that are not solvable without the paper context**, we had a sense of (a) **human baselines** to verify the validity of those problems, and (b) **frontier model evaluations** to see headroom on this set.

**Dataset Code Accessibility:**

Yes

**Dataset Code Comments:**

The data looks quite accessible and there is some nice tooling developed by the authors for contributing tests.

**Ethical Considerations:**

No, there are no or only very minor ethics concerns

**Final Justification:**

The reviewer addressed the main issue I had regarding the possibility of question ambiguity with a human baseline. They also address issues regarding a scaled pass rate that make it difficult to parse the capability of current models.

They did not resolve concerns of whether a model with some scaffolding (ability to iterate on errors) can almost solve the benchmark.

**Limitations Weaknesses:**

It is unclear whether or not frontier models may be close to solving the benchmark, especially inside additional agent harnesses or given some error feedback (e.g. public tests). This may diminish some of the impact of benchmark. Elaborating further:

 - Although authors claim the best models “correctly implement less than 40% of the code”,  I am skeptical of their derivation of this via the  “scaled pass rate”. This metric is nonstandard and introduces additional complexity to the benchmark. If one wants to highlight the importance of solving the more difficult problems (c.f. “[regular pass rate] treats all snippets equally, regardless of their complexity or size…”), a more natural approach is to introduce a hard subset (e.g. ResearchCodeBench-HARD, in the spirit of MATH500 vs MATH).

- With this in mind, I am interested in human performance (or pass@n rates) to contextualize the current results: how much of the remaining mistakes produced by frontier models are due to a lack of capability or rather, ambiguity between the test statement and acceptance criteria?

A significant amount of the problems are solved without using the provided paper as context. This suggests that these specific problems are not measuring the models’ ability to implement novel machine learning research code.

**Strengths Contributions:**

Understanding how good LLMs are at implementing new research ideas is obviously an important and interesting problem.

The writing of the paper is very good. The mechanics and design principles of the benchmark are clearly explained.

The design of the benchmark is thoughtful. Indeed the paper identifies the tradeoff between (1) adding enough structure to the problem so solutions are appropriately clear and so the results can be verified accurately and cheaply (i.e. with unit tests); and (2) keeping in a sufficient amount of “messiness” so results on the benchmark can simulate real world model usage. I think the benchmark positions itself into a useful niche of being fast and reliable to evaluate while staying relatively grounded in real world problems.

The breadth of the empirical study is good. They test a lot of models and give some good analysis regarding contamination and error.

---

> ### Author Rebuttal · Authors · 2025-07-31
>
> **Reviewer Comment:**
> I am skeptical of the "scaled pass rate" metric. A more natural approach would be to introduce a hard subset (e.g. ResearchCodeBench-HARD, like MATH500 vs MATH).
>
> **Author Response:**
> We thank the reviewer for this insightful suggestion. We agree that introducing a ResearchCodeBench-HARD subset provides a cleaner and more interpretable approach than our scaled pass rate metric. Following the reviewer's recommendation, we have created ResearchCodeBench-HARD by selecting the 109 most challenging tasks (bottom 50% by pass rate) from our benchmark.
>
> The best-performing model achieves only 33.0% pass rate on ResearchCodeBench-HARD, validating our claim using standard metrics. Notably, 43 tasks have 0% pass rate across all 32 models, including GPS reparameterization (Zhang et al., 2025), differential attention (Ye et al., 2025), and dynamic programming (Pei et al., 2025).
>
> **Table 1: ResearchCodeBench-HARD Performance (Top 10 + Selected Bottom)**
>
> | Rank | Model | Pass Rate |
> |------|-------|-----------|
> | 1 | Gemini-2.5-Pro-Preview-05-06 | 33.0% |
> | 2 | O4-Mini (High) | 28.4% |
> | 3 | O3 (High) | 25.7% |
> | 4 | Gemini-2.5-Pro-Preview-03-25 | 25.7% |
> | 5 | O3-Mini (High) | 18.3% |
> | 6 | GPT-4.1 | 16.5% |
> | 7 | O1 (High) | 14.7% |
> | 8 | Gemini-2.5-Flash-Preview-04-17 | 14.7% |
> | 9 | Claude-3.5-Sonnet | 13.8% |
> | 10 | Claude-3.7-Sonnet | 13.8% |
> | ... | ... | ... |
> | 29-32 | 4 models (Llama-3.3-70B, etc.) | 0.0% |
>
> The HARD subset includes tasks from mathematical algorithms (GPS, conformal isometry), deep learning (differential attention, masked attention), optimization (dynamic programming, lazy Newton), and probabilistic methods (importance sampling, Gumbel transforms). We will replace scaled pass rates with ResearchCodeBench-HARD results throughout the paper.
>
>
>
> ---
>
> **Reviewer Comment:**
> It is unclear whether or not frontier models may be close to solving the benchmark, especially inside additional agent harnesses or given some error feedback (e.g. public tests).
>
> **Author Response:**
> This is an excellent point about the potential for agent-based approaches to significantly improve performance. We acknowledge that our current evaluation represents a lower bound on frontier model capabilities, as we evaluate models in isolation without iterative feedback or tool access.
>
> **Future work on agentic evaluation** would be valuable and represents a natural extension of our benchmark:
>
> 1. **Iterative refinement**: Provide runtime error messages and stack traces to enable debugging while preserving the novel implementation challenge.
>
> 2. **Tool-augmented search**: Allow access to documentation, code search, and computational tools while respecting knowledge cutoffs. The key challenge is designing this to test research implementation ability rather than information retrieval.
>
> However, we believe our single-turn evaluation provides important baseline insights. Even with agentic approaches, the core challenge remains: can models understand and implement novel research contributions? Our results suggest substantial room for improvement even without agentic tools, as many tasks require deep algorithmic understanding that iterative refinement alone may not address.
>
> We will add this important limitation and future work discussion to the main paper.
>
> ---
>
> **Reviewer Comment:**
> I am interested in human performance to contextualize the results: are remaining mistakes due to lack of capability or task ambiguity?
>
> **Author Response:**
> Thank you for raising this important point about human performance benchmarking. We agree this would provide valuable context for interpreting model results. During the rebuttal period, we conducted a preliminary human study with one expert participant familiar with the selected research areas. The results are illuminating:
>
> **Table: Human Expert Performance on ResearchCodeBench (Preliminary Study)**
>
> | Paper | Human Performance | Total Tasks |
> |-------|-------------------|-------------|
> | advantage-alignment (Duque et al., 2025) | 3/5 (60%) | 5 |
> | Diff-Transformer (Ye et al., 2025) | 5/7 (71%) | 7 |
> | DiffusionDPO (Wallace et al., 2024) | 9/9 (100%) | 9 |
> | DyT (Zhu et al., 2025) | 9/9 (100%) | 9 |
> | GPS (Zhang et al., 2025) | 6/6 (100%) | 6 |
> | eomt (Kerssies et al., 2025) | 30/37 (81%) | 37 |
> | hyla (Schug et al., 2025) | 2/3 (67%) | 3 |
> | llm-sci-use (Liang et al., 2024) | 6/15 (40%) | 15 |
> | Tanh-Init (Lee et al., 2024) | 4/4 (100%) | 4 |
> | minp (Nguyen et al., 2024) | 5/7 (71%) | 7 |
> | **Overall** | **79/102 (77.5%)** | **102** |
>
> Comparing this with LLM performance on the same 10 papers reveals important insights:
>
> **Comparative Performance (Same 10 Papers, 102 Tasks, Standard Pass@1):**
> - **Human expert**: 77.5% (79/102)
> - **Best LLM** (Gemini-2.5-Pro-Preview-05-06): 68.6% (70/102)
> - **Performance gap**: 8.9 percentage points
>
> **Table: Top 5 LLMs vs Human Performance (Same 10 Papers)**
>
> | Model | Performance | Gap vs Human |
> |-------|-------------|--------------|
> | Human Expert | 77.5% (79/102) | - |
> | Gemini-2.5-Pro-Preview-05-06 | 68.6% (70/102) | -8.9% |
> | O3 (High) | 66.7% (68/102) | -10.8% |
> | Gemini-2.5-Pro-Preview-03-25 | 65.7% (67/102) | -11.8% |
> | Claude-3.7-Sonnet | 60.8% (62/102) | -16.7% |
> | O4-Mini (High) | 59.8% (61/102) | -17.7% |
>
> *Note: All metrics use standard pass@1 rates (tasks passed / total tasks), not scaled pass rates.*
>
> Key findings: (1) The 8.9pp gap suggests significant headroom, (2) Humans excel at mathematical tasks (GPS: 100% vs 17%) while LLMs handle data processing better (llm-sci-use: 40% vs 80%), (3) Complementary strengths indicate different problem-solving approaches.
>
> We acknowledge limitations: humans can have access to all published work before test while LLMs have controlled cutoffs, and recruiting domain experts is challenging. Despite n=1, this provides valuable calibration showing significant headroom.
>
> ---
>
> **Reviewer Comment:**
> Many problems are solved without paper context, suggesting they don't measure novel ML research implementation ability.
>
> **Author Response:**
> We acknowledge this important observation and have conducted additional analysis to address this concern. While 30.2% of tasks achieve >50% success rate without paper context across all 32 evaluated models, we contend that this does not diminish the benchmark's validity for two key reasons:
>
> **1. Significant portion requires paper context:** Our analysis reveals that **28.3% of tasks (60/212) are completely impossible without paper context**—no model among all 32 evaluated LLMs can solve them. These tasks see dramatic improvements with paper access (e.g., Tanh-Init weight update: 0% → 68.8%, schedule-free gradient interpolation: 0% → 46.9%). This demonstrates that nearly a third of our benchmark specifically tests novel research implementation abilities.
>
> **2. Contextual reasoning in research settings:** Even "paper-independent" tasks require models to implement novel research code through reasoning from available contextual information—comments describing research intent, function signatures indicating novel interfaces, and surrounding code revealing architectural patterns. This contextual implementation reasoning mirrors realistic development scenarios where AI assistants provide auto-completion or code suggestions based on local context rather than full paper access—a common use case in research development workflows.
>
> ResearchCodeBench evaluates a realistic spectrum of research implementation challenges, from deep paper understanding to contextual reasoning abilities essential for AI-assisted research development. This design captures the complexity of implementing cutting-edge ML research.
>
> ---
>
> **Reviewer Comment:**
> Models with earlier cutoffs (e.g. OpenAI) should show smaller contamination drops than newer models (e.g. Gemini), but this isn't observed. Why?
>
> **Author Response:**
> You raise an excellent point about the expected correlation between training cutoff dates and performance drops on contamination-free data. This observation prompted us to examine our results more carefully, and we found some counterintuitive patterns that suggest a more complex interaction between contamination, model capabilities, and task difficulty.
>
> **Empirical observations:** While we do see general contamination effects (all models drop on contamination-free subset), the magnitude doesn't always correlate with cutoff dates as expected. For example:
> - Models with 2023 cutoffs sometimes show larger drops than models with 2024-2025 cutoffs
> - Some newer models maintain relatively stable performance on contamination-free tasks
>
> **Potential explanations for this pattern:**
>
> 1. **Capability-contamination interaction**: Newer models often have stronger general reasoning abilities that help them tackle novel tasks, partially offsetting the loss of contamination advantage. Older models may rely more heavily on memorization, leading to larger drops when that advantage is removed.
>
> 2. **Contamination-free subset composition**: Our contamination-free subset may inadvertently contain a higher proportion of tasks requiring advanced capabilities (e.g., complex mathematical reasoning, novel architectural patterns). This would disproportionately impact older models regardless of contamination.
>
> 3. **Training data diversity**: Models trained on more diverse codebases (often newer models) may generalize better to unseen research code patterns, while older models might be more specialized to their training distribution.
>
> We will add analysis correlating cutoff dates with performance drops, control for task difficulty, and report model-specific impacts grouped by cutoff date. This will help disambiguate whether patterns stem from capability differences or task selection bias.

---

> > ### Comment · Reviewer_6kcd · 2025-08-06
> > **Thank you for the new results**
> >
> > I thank the authors for the comprehensive rebuttal that has addressed my concerns, particularly the human study to understand headroom and addressing concerns regarding the scaled pass rate.
> >
> > I trust the authors will communicate more details about the human study design (how the participant was selected, level of experience, time allotted, whether they got compiler and test feedback) in the final manuscript. It would also be great to know the human performance on the hard subset.
> >
> > This study is very exciting work, and I recommend acceptance (raised my score).

---

### Decision · Program_Chairs · 2025-09-18

**Decision:**

Accept (spotlight)

**Comment:**

The paper proposes a new benchmark for LLM driven ML research on implementing novel machine learning algorithms based on reference papers. The benchmark uses fill-in-the-blank questions to focus on critical research aspects instead of the larger codebase design.
The paper provides a clear definition of benchmark criteria, an automated evaluation strategy based on testing and a rigorous evaluation of current models. While potential agent-based solutions are not investigated in this work, the benchmark is well-designed and addresses a clear research need.